# Using antibodies to control DNA-templated chemical reactions

Lorena Baranda Pellejero [1], Malihe Mahdifar[1], Gianfranco Ercolani [1], Jonathan Watson[2], Tom Brown Jr [2] & Francesco Ricci [1✉]

DNA-templated synthesis takes advantage of the programmability of DNA-DNA interactions to accelerate chemical reactions under diluted conditions upon sequence-specific hybridization. While this strategy has proven advantageous for a variety of applications, including sensing and drug discovery, it has been so far limited to the use of nucleic acids as templating elements. Here, we report the rational design of DNA templated synthesis controlled by specific IgG antibodies. Our approach is based on the co-localization of reactants induced by the bivalent binding of a specific IgG antibody to two antigen-conjugated DNA templating strands that triggers a chemical reaction that would be otherwise too slow under diluted conditions. This strategy is versatile, orthogonal and adaptable to different IgG antibodies and can be employed to achieve the targeted synthesis of clinically-relevant molecules in the presence of specific IgG biomarker antibodies.

[1] Chemistry Department, University of Rome, Tor Vergata, Via della Ricerca Scientifica, 00133 Rome, Italy. [2] ATDBio Ltd, Magdalen Centre, Oxford Science Park, Robert Robinson Avenue, Oxford OX4 4GA, UK. ✉email: francesco.ricci@uniroma2.it

In the crowded cellular environment, where thousands of different species coexist in a limited volume, it is crucial to control chemical reactivity in a highly precise manner so that nonspecific reactions that might lead to undesired effects are avoided[1]. Faced with this challenge, Nature has evolved mechanisms that allow the co-localization of biomolecules in a confined volume, which results in an increase of their effective local concentrations[2,3]. Such local concentration enhancement triggers intermolecular reactions that would otherwise be unlikely to occur at the low concentrations found in cells[4,5].

Inspired by this mechanism, the compartmentalization of reactive species into a confined nanoscale space has been artificially recreated in synthetic systems using different approaches[6–9]. Artificial molecular containers, also defined as "molecular flasks"[10], that are able to modulate the chemical reactivity through confinement, include molecular capsules and boxes[11,12], zeolites[13], covalent organic frameworks[14], and metal–organic frameworks[15,16]. Another approach to enhance the reaction rate between reactive species by increasing their effective concentration is based on the use of templates, molecular scaffolds designed to co-localize, and orient reactive units in a confined volume. Such templated spatial arrangements not only increase the effective concentration through proximity effects but also specifically orient the reactive groups to enable reactions in a target-specific fashion[17,18]. One of the most versatile and flexible examples of this is DNA templated synthesis (DTS), in which the reactive groups are conjugated to synthetic oligonucleotide sequences (DNA or RNA). The sequence-specific hybridization of these modified-sequences leads to co-localization of the reactive groups thus enabling chemical reactions under highly dilute conditions. DTS benefits from the high predictability of Watson–Crick interactions, the low cost of synthesis, and the ease of attaching different reactive groups to synthetic DNA oligonucleotides[19]. The range of chemistries compatible with this method has gradually expanded over the years, and a variety of chemical reactions can now be controlled simultaneously, yielding a wide range of synthetic molecules structurally unrelated to nucleic acids[20–22]. Numerous applications of DTS have been reported, ranging from nucleic acid detection[23–28] to drug-release[29,30] and small-molecule drug discovery[22,31]. For this latter application, the selectivity and specificity encoded in each nucleic acid strand has enabled the creation of large combinatorial DNA-encoded libraries of reactions in a single solution and has allowed the exploration of much wider chemical spaces when compared with traditional high-throughput screening methods[32]. While the above examples clearly demonstrate the advantages of DTS, additional features might help in improving the utility of this approach, leading to new practical applications. A critical limitation of DTS is that DNA-templated reactions rely solely on nucleic acids as templating agents (and therefore on Watson–Crick interactions). This might ultimately limit the possible applications of DTS in more diverse research fields. For example, triggering DNA-templated reactions with other non-nucleic acid co-templating biomolecules, including relevant clinical biomarkers, would increase both the versatility and utility of DTS and broaden the contexts in which chemical reactivity could be controlled.

Motivated by the above arguments, we demonstrate here a strategy for the control of DNA-templated chemical reactions using specific IgG antibodies as co-templating agents. This approach takes advantage of both the bivalent binding of IgG antibodies and the possibility of using nucleic acids as versatile scaffolds to conjugate reactive groups and different recognition molecules. This will ultimately enable synthetic antibody (Ab)-directed chemical reactions, that might find future applications in the fields of clinical diagnosis and drug-delivery.

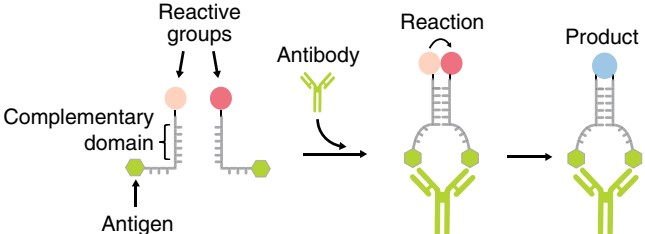

**Fig. 1 Antibody-templated reactions mediated by DNA hybridization.** Two complementary DNA oligonucleotides are each conjugated at one end with a reactive group and at the other end with an antigen. At low oligonucleotide concentrations (nM) duplex formation is not favored and the reaction rate between the two reactive groups is negligible. Bivalent binding of the specific antibody to the two antigen-conjugated oligonucleotides induces an increase of their effective concentration thus leading to the efficient duplex formation. The proximity effect induced by antibody-mediated duplex formation considerably enhances the rate of the reaction between the reactive groups.

## Results

**Design of Ab-directed DNA-templated synthesis.** Our strategy to achieve Ab-directed DNA-templated synthesis takes advantage of the following features: first, the Y-shaped geometry that all IgG antibodies share, with two identical binding sites separated by about 6–14 nm[33–35]; second, the possibility to easily conjugate different recognition elements to synthetic nucleic acid strands[36]. More specifically, we conjugated a reactive group and a recognition element (i.e., antigen) to each of a pair of templating synthetic oligonucleotides (Fig. 1). The two oligonucleotides have complementary domains designed to hybridize and form a duplex that is however unstable under the experimental conditions employed (low nM concentration). As a consequence, the reaction rate between the reactive groups linked to the oligonucleotides is negligible. Upon Ab binding, the two oligonucleotides are co-localized into a confined volume thus inducing efficient hybridization and stable duplex formation. Such Ab-mediated hybridization ultimately brings the two reactive groups into close proximity, greatly increasing their effective concentration and accelerating the resulting reaction (Fig. 1).

**Thermodynamic characterization of the Ab-induced co-localization.** Instrumental for this strategy to observe Ab-directed DNA-templated reaction is the need to find an optimal thermodynamic trade-off so that the two templating oligonucleotides bearing the reactive groups efficiently hybridize only upon bivalent Ab binding. To identify such optimal trade-off, we designed a set of templating oligonucleotide pairs differing in the length of their complementary domains (and, therefore, duplex stabilities). More specifically, we designed complementary domains with lengths ranging from 6 to 16 nucleotides (nt), corresponding to predicted duplex free energy values from $-10.3$ to $-28.5$ kcal/mol, respectively (Fig. 2a)[37]. For thermodynamic characterization of the duplex formation, we initially labeled the two ends of the templating pairs with a FRET couple in place of the reactive groups so that duplex formation leads to a fluorescence signal decrease (Fig. 2b). Also, as a first step towards the characterization of the Ab-induced co-localization, we engineered a bivalent DNA strand designed to act as an Ab-mimic and bind the tails of the templating strands in a way similar to that expected from an IgG Ab (Fig. 2a). This Ab-mimic provided a system to study the duplex formation and successive reaction induced by the binding of a bivalent DNA co-template agent without side reactions that could occur owing to the presence of an Ab in the reaction solution or interference from the Ab-antigen recognition event.

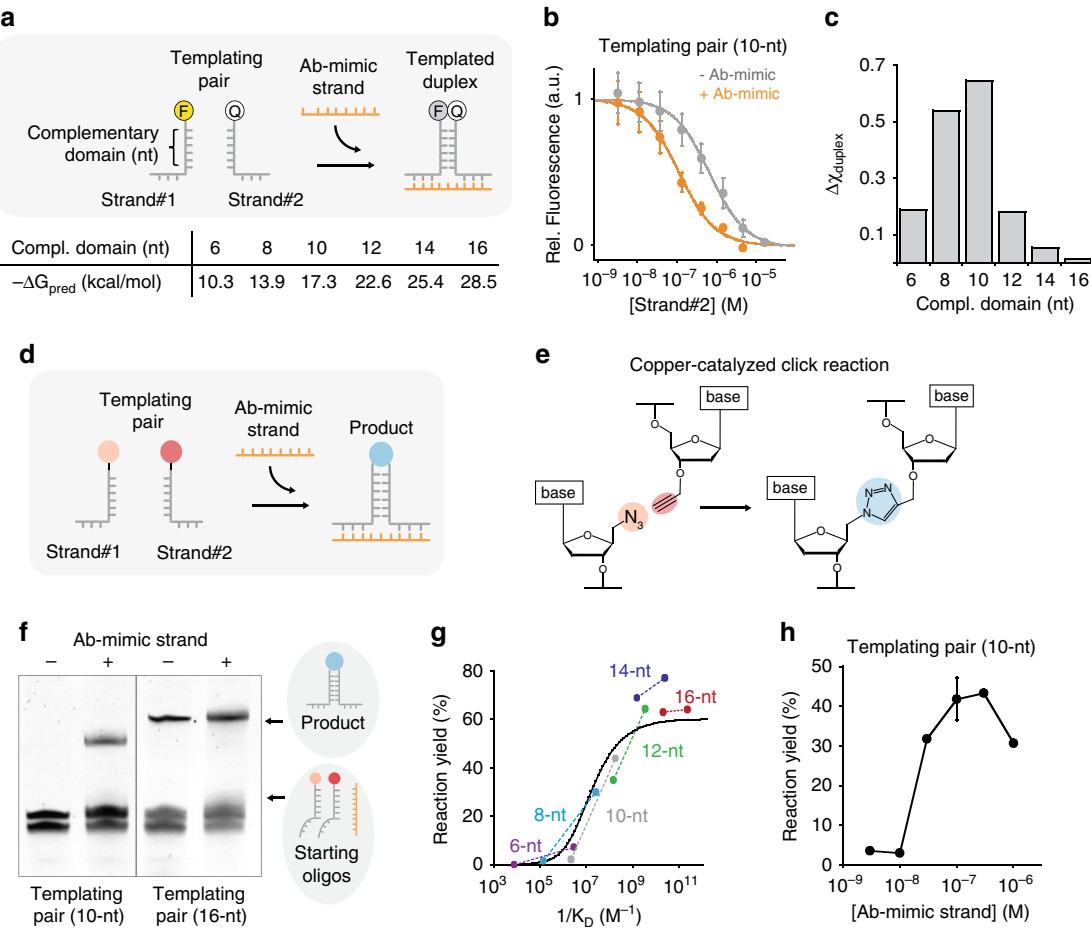

**Fig. 2 Reaction templated by an antibody-mimic (Ab-mimic) DNA strand. a** We designed a DNA oligonucleotide that mimics the co-localization effect induced by the bivalent binding of an IgG antibody to a templating pair (see Fig. 1). Optically modified oligonucleotides were initially tested to study the effect of the Ab-mimic on the duplex formation. Templating pairs with complementary domains of different lengths were used. Shown are the predicted free energy values of duplex formation between the templating pair. **b** Binding curves between Strand#1 and Strand#2 (the complementary domain of 10-nt) in the absence and presence of the Ab-mimic strand. **c** Difference in duplex molar fraction for different lengths of the complementary domain in the presence and absence of Ab-mimic strand calculated for an equimolar concentration of templating pairs and Ab-mimic strand of 100 nM. Data from Supplementary Table 4. **d** Scheme depicting the reaction co-templated by the Ab-mimic strand. **e** Azide and alkyne groups leading to copper-catalyzed click (CuAAC) reaction were used as reactive groups. **f** Denaturing PAGE of CuAAC reaction in the presence and absence of the Ab-mimic for templating pairs with complementary domains of 10 and 16 nt. **g** Plot of reaction yield vs. the inverse of dissociation constant. Each pair of connected dots is relative to the indicated length of complementary domains. For each pair, the dot on the left corresponds to the experimental yield in the absence of the Ab-mimic strand vs. $1/K_{D\_no\_template}$, whereas that on the right corresponds to the experimental yield in the presence of the Ab-mimic strand vs. $1/K_{D\_Ab\_mimic}$. The solid curve is the theoretical yield obtained by fitting the experimental points to the kinetic model of the system. **h** Reaction yield determined by densitometric analysis of PAGE bands at different concentrations of Ab-mimic strand (10-nt templating pair). The fluorescence experiments (panel **b**) were performed at 37 °C in 25 mM HEPES buffer pH 7.2, 0.1 M NaCl at a fixed concentration of Strand#1 (10 nM) and increasing concentrations of Strand#2 in the absence and presence (i.e., 100 nM) of Ab-mimic strand. Templated CuAAC reactions (panel **f**) were performed at 37 °C for 2 h at equimolar concentrations (i.e., 100 nM) of Strand#1, Strand#2 and Ab-mimic (when indicated) in presence of Cu(I) catalyst (details of catalyst composition described in methods part) in 25 mM HEPES buffer pH 7.2, 0.1 M NaCl. The experimental values represent averages of three separate measurements and the error bars reflect the standard deviations. Source data are provided as a Source Data file.

We then obtained binding curves for the two templating oligonucleotides in the absence and presence of a fixed concentration of the Ab-mimic strand (Fig. 2b shows the results for the 10-nt templating pairs; see Supplementary Figs. 1–5 for the other pairs). The resulting curves allowed us to either evaluate or estimate (with some additional assumptions) the values of the dissociation constants between templating strands in the absence ($K_{D\_no\_template}$) and presence of a given concentration of the Ab-mimic strand ($K_{D\_Ab\_mimic}$), and to derive a full thermodynamic understanding of the system (see "Methods" and SI for a full discussion of dissociation constants evaluation and the estimated

values). As expected, co-localization induced by the bivalent binding of the Ab-mimic strand led to an increase in affinity between strand #1 and strand #2 that was dependent on the concentration of the Ab-mimic strand. Using the determined dissociation constants we could estimate the duplex molar fraction values in the absence and presence of the Ab-mimic strand at the conditions that would be used for subsequent templated reactions (i.e., the equimolar concentration of templating pairs, and Ab-mimic strand, 100 nM) (Supplementary Fig. 6). We found that a templating pair with a 10-nt complementary domain led to the highest increase in duplex molar fraction (Fig. 2c).

**Reaction templated by an Ab-mimic DNA strand**. We then moved to demonstrate that the same 10-nt complementary templating pairs could lead to efficient DNA-templated synthesis mediated by the presence of the Ab-mimic strand (Fig. 2d). We initially employed a classic biorthogonal chemical reaction: the copper(I)-catalyzed azide-alkyne cycloaddition (CuAAC) reaction (Fig. 2e). Click reactions were preferred owing to the selectivity offered by the reactants and the efficiency of the reaction[38,39]. The two reactive groups were thus conjugated to the 5′ and 3′-ends of two complementary DNA oligonucleotides. We observed that the DNA-templated CuAAC reaction was triggered only in the presence of the Ab-mimic strand. A visible band was obtained in the presence of the Ab-mimic strand while no product was observed by gel electrophoresis in its absence (Fig. 2f, mass spectrometry (MS) analysis of product shown in Supplementary Fig. 7). Of note, the same experiment carried out with longer duplex-forming domains (i.e., 16 nt) showed, as expected, comparable reaction efficiency in the absence and presence of the Ab-mimic strand (Fig. 2f).

To further demonstrate the proposed mechanism, we performed templated reactions with azide- and alkyne-modified oligonucleotides, with the duplex-forming domains of different lengths tested above. Reaction yields in the presence and absence of Ab-mimic strand are represented as a pair of connected dots for each complementary length (Fig. 2g). The presence of the Ab-mimic strand led to a marginal increase in product yield (measured by the intensity of the product gel-electrophoresis band) with the longer duplex regions (i.e., 16 and 14 nt). Similar behavior was also observed with the shortest complementary domain (i.e., 6 nt). Conversely, the Ab-mimic strand increased product yield for the templating pairs of intermediate length (i.e., 8, 10, and 12 nt). We devised a kinetic model that, given the initial reactant concentrations and reaction time, predicts the product yield as a function of the kinetic constant for the reaction between the co-localized end groups and the dissociation constant of the templating strands (see Methods and SI for a full discussion of the kinetic model). The theoretical curve predicted by the model as a function of $1/K_D$ is shown in Fig. 2g. It was calculated by optimizing the kinetic constant so as to give the best fit to the experimental yields. The curve, with a single adjustable parameter, does an excellent job of fitting the data over a span of about 8 powers of 10 for the values of the dissociation constants.

Templated-induced product formation is concentration-dependent. To demonstrate this, we performed experiments at different concentrations of the Ab-mimic strand finding a concentration-dependent increase in product yield up to a maximum of 100 nM of Ab-mimic strand, which corresponds to the concentration of the reactive groups-conjugated strands in solution (Fig. 2h). At higher concentrations of Ab-mimic strand (i.e., 1 μM) a lower product yield was observed, probably due to the dehybridization of the template duplex induced by the excess of Ab-mimic strand used (Supplementary Fig. 8); an effect also known as "combinatorial inhibition"[40].

**Antibodies as co-templating agents for chemical reactions**. As a first testbed to demonstrate Ab-templated synthesis we initially employed as recognition element (i.e., antigen) the small-molecule hapten digoxigenin (DIG), and we used the anti-DIG Ab as co-templating IgG Ab (Fig. 3a). We slightly re-engineered our system to simplify the synthesis of reactant strands. More specifically, we redesigned one of the two duplex-forming templating strands to contain a 17-nt tail that hybridizes to a complementary oligonucleotide strand linked to the recognition element (i.e., antigen) (Fig. 3a). We also used a set of optically labeled oligonucleotides of different length to perform a

thermodynamic characterization of the Ab-induced co-localization mechanism (Supplementary Fig. 9). The results confirm the conclusions reached with the Ab-mimic strand previously described: bivalent Ab binding induced an increase in the binding affinity between the templating oligonucleotides (Supplementary Figs. 10–15). Also in this case we found that a 10-nt templating pair provided the strongest difference in hybridization efficiency between the absence and presence of the anti-DIG Ab (Supplementary Figs. 16 and 17). We then demonstrated an efficient Ab-induced templated CuAAC reaction between an azide and an alkyne group present in the 10-nt templating pair (Fig. 3b). While no product was observed by gel electrophoresis experiments in the absence of anti-DIG Ab, the reaction carried out with anti-DIG Ab (300 nM) led to a strong visible band corresponding to the expected product (Fig. 3c). The same experiment carried out with longer duplex forming domains (i.e., 16 nt) showed, as expected, no significant difference in product yield in the absence and presence of anti-DIG Ab (Fig. 3c). Electrospray ionization (ESI)–MS experiments found the expected masses of the products formed by anti-DIG Ab co-localization (Fig. 3d, top) and by a control experiment carried out with the templating pair of 16 nt in the absence of anti-DIG Ab (Fig. 3d, bottom).

The proposed mechanism is further supported by the results obtained with the templating pairs of different lengths (Fig. 3e). For longer duplex domains (16 and 14 nt), we observed similar product yields both in the absence and presence of the anti-DIG Ab, whereas for intermediate duplex-forming domains (12 and 10 nt), product formation only occurred in the presence of anti-DIG Ab. The same kinetic model used to fit the data in the absence and in the presence of the Ab-mimic strand successfully fits the experimental data in the absence and in the presence of the anti-DIG Ab. Indeed, the curve shown in Fig. 3e is identical to that shown in Fig. 2g. The Ab-templated CuAAC reaction is also concentration-dependent, and we observed increasing yields of product formation at increasing concentrations of anti-DIG Ab, with a maximum reached at 300 nM of anti-DIG (Fig. 3f). Additional support for the proposed Ab-templated mechanism comes from different control experiments (Fig. 3g). For example, we performed the same Ab-templated reaction described above (using the 10-nt templating pair) in the presence of anti-DIG Fab fragment (which contains only a single DIG binding site); as expected, no product was observed (Fig. 3g). In another experiment, we used a saturating concentration of DIG which, by competing with the templating strands for the binding of anti-DIG Ab, gave no visible product when analyzed by polyacrylamide gel electrophoresis (PAGE) (Fig. 3g). We also found that no templated product was observed using saturating concentrations of a different nonspecific Ab (anti-DNP) (Fig. 3g). These results further support the hypothesis that specific bivalent Ab binding is necessary to accelerate the reaction and to observe product formation under the diluted concentrations employed.

The modular nature of this approach allows an easy generalization to other templated reactions. To demonstrate this, we redesigned the system to template a phosphoramidate ligation[41] (Fig. 3h), which, as opposed to the previous CuAAC reaction, is not strictly biorthogonal. Amine- and phosphate-modified strands were ligated following the same approach previously described (Fig. 3h). We first confirmed that this reaction can be accelerated by bivalent binding of an Ab-mimic strand to the 10-nt templating pair (Supplementary Fig. 18). We then moved to the use of anti-DIG Ab and demonstrated that its presence is crucial for efficient phosphoramidate ligation: we observed product formation only when anti-DIG Ab was present in the reaction mixture (10-nt templating pair, Fig. 3i). Moreover, control experiments with the 16-nt templating pair showed, as expected, no significant difference in product yield (Fig. 3i).

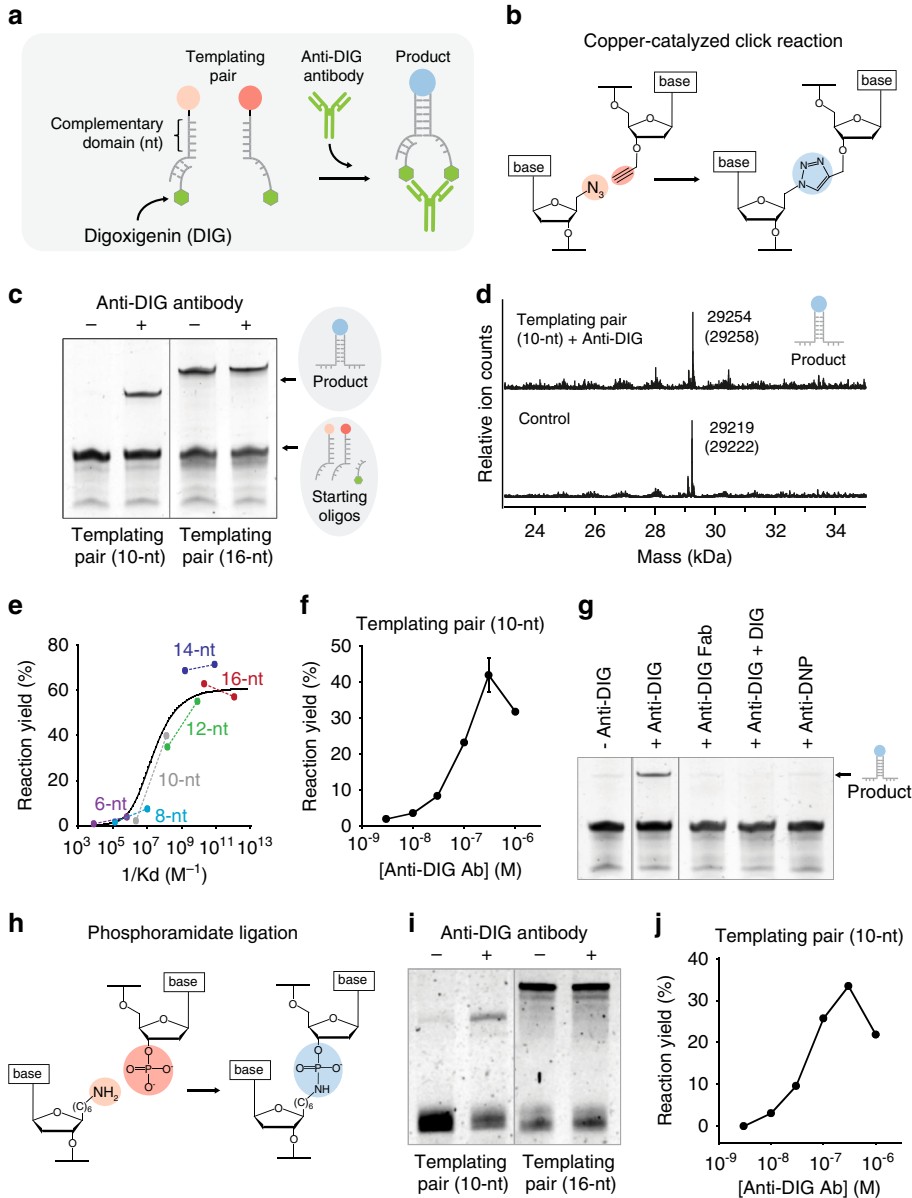

**Fig. 3 Antibodies as co-templating agents for chemical reactions. a** Scheme of the reaction triggered by anti-DIG antibody. The templating pairs were modified with the reactive groups and with an antigen (green hexagon, here digoxigenin, DIG). **b** Azide and alkyne groups leading to CuAAC reaction were used as reactive groups. **c** Denaturing PAGE of CuAAC reactions in the absence and presence of anti-DIG antibody for complementary domains of 10 and 16 nt. **d** ESI(−) mass spectra of the reaction in the presence and absence (control) of the anti-DIG antibody for complementary domains of 10 and 16 nt, respectively, allowing identification of the product. Observed and expected (in brackets) m/z values are indicated. **e** Plot of reaction yield vs. the inverse of dissociation constant. Each pair of connected dots is relative to the indicated length of complementary domains. For each pair, the dot on the left corresponds to the experimental yield in the absence of the anti-DIG antibody vs. $1/K_{D\_no\_template}$, whereas that on the right corresponds to the experimental yield in the presence of anti-DIG antibody vs. $1/K_{D\_Anti\_DIG\_Ab}$. The solid curve is the theoretical yield obtained by fitting the experimental points to the kinetic model of the system. **f** Reaction yield determined by densitometric analysis of PAGE bands at different concentrations of anti-DIG antibody (with 10-nt templating pair). **g** Denaturing PAGE of CuAAC reactions carried out in the absence of co-templating agent (first lane) or in the presence of anti-DIG antibody (lane 2), monovalent anti-DIG Fab fragment (lane 3), anti-DIG antibody and saturating concentration of free DIG (1 μM) in solution (lane 4) and a nonspecific antibody (anti-DNP) (lane 5). **h** Scheme of phosphoramidate ligation triggered by the anti-DIG antibody. In this case, amine and phosphate groups were used as reactive groups. **i** Denaturing PAGE of phosphoramidate reactions in the absence and presence of anti-DIG antibody for complementary domains of 10 and 16 nt. **j** Reaction yield determined by densitometric analysis of PAGE bands at different concentrations of anti-DIG antibody (with 10-nt templating pair). Templated CuAAC reactions in this figure were all performed at 37 °C for 2 h at a 100 nM concentration of templating strands and 300 nM of the corresponding antibody in presence of Cu(I) catalyst in 25 mM HEPES buffer pH 7.2, 0.1 M NaCl. Templated phosphoramidate reactions were performed at 37 °C for 2 h at 100 nM concentration of templating strands and 300 nM of anti-DIG antibody in presence of 25 mM EDC.HCl and 100 mM 1-(2-hydroxyethyl) imidazole in 25 mM HEPES buffer pH 7.2, 0.1 M NaCl. The experimental values represent averages of three separate measurements and the error bars reflect the standard deviations. Source data are provided as a Source Data file.

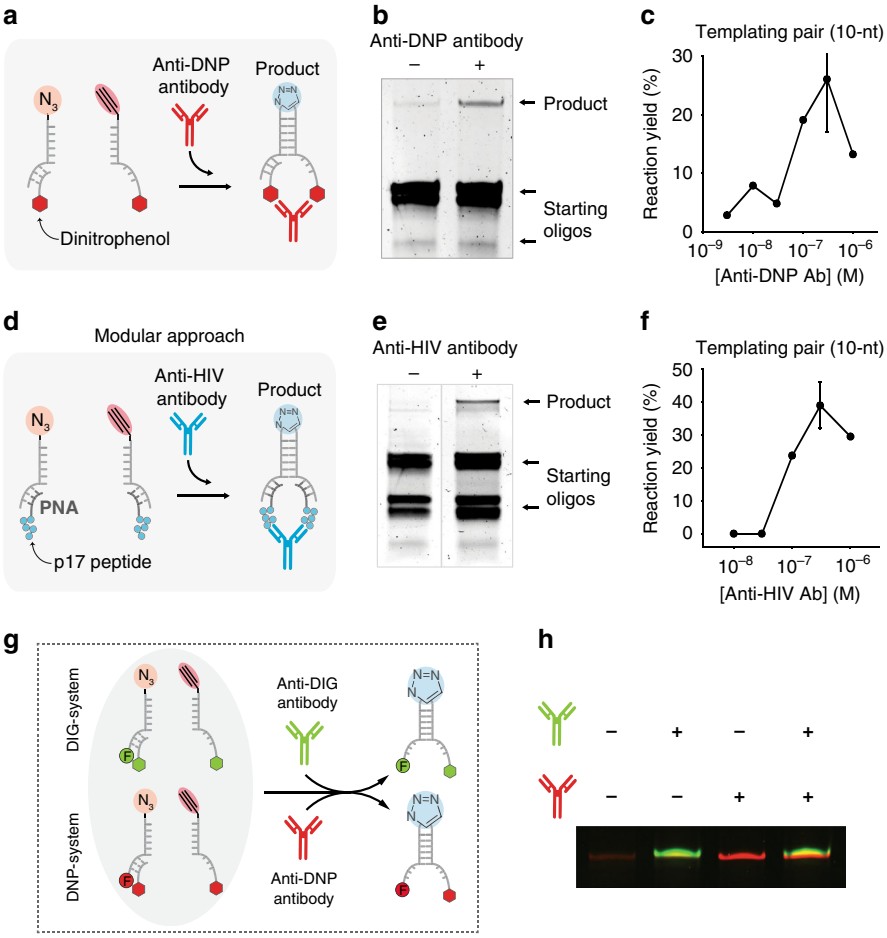

**Fig. 4 Orthogonal control of reactions mediated by different specific antibodies. a** Scheme of the CuAAC reaction triggered by an anti-DNP antibody. The templating pairs were modified with azide and alkyne reactive groups and with DNP. **b** Denaturing PAGE of CuAAC reactions in the absence and presence of anti-DNP antibody for the 10-nt complementary domain. **c** Reaction yield determined by densitometric analysis of PAGE bands at different concentrations of anti-DNP antibody (10-nt templating pair). **d** Scheme of the modular approach used to achieve CuAAC reaction triggered by an anti-HIV antibody. The templating pairs were modified with azide and alkyne reactive groups. The templating pairs were both hybridized to strands conjugated at one end with a p17 peptide recognized by an anti-HIV antibody. **e** Denaturing PAGE of CuAAC reactions in the absence and presence of anti-HIV antibody for the 10-nt complementary domain. **f** Reaction yield determined by densitometric analysis of PAGE bands at different concentrations of anti-HIV antibody (10-nt templating pair). **g** Two systems with templating pairs each labeled with a specific recognition element (DIG and DNP) were used in the same solution. One strand from each templating pair was also labeled with a different fluorophore to differentiate the products. **h** Denaturing PAGE of CuAAC reactions in the absence and presence of the two specific antibodies. Templated CuAAC reactions in this figure were all performed at 37 °C for 2 h at a 100 nM concentration of templating strands and 300 nM of the specific antibody in presence of Cu(I) catalyst in 25 mM HEPES buffer pH 7.2, 0.1 M NaCl. The experimental values represent averages of two separate measurements and the error bars reflect the standard deviations. Source data are provided as a Source Data file.

Reaction yield correlates, once again, with Ab concentration, and the maximum yield was obtained at 300 nM Ab concentration, as with the previous reaction (Fig. 3j).

Our Ab-templated reaction is in principle generalizable and adaptable to other IgG antibodies via changing the recognition element. To demonstrate this, we re-engineered a second Ab-templated CuAAC reaction by designing a new pair of templating strands conjugated with a different antigen (i.e., dinitrophenol, DNP) (Fig. 4a). We found that the templated reaction could occur only in the presence of anti-DNP Ab (Fig. 4b, MS analysis of product shown in Supplementary Fig. 19). This new reaction had efficiency and kinetics comparable to those observed with the anti-DIG templated reaction (Fig. 4c). In order to demonstrate the applicability of this strategy to more clinically relevant IgG antibodies, the Ab-induced DNA templated design was further re-engineered so that more complex recognition elements (such

as peptides) could be employed (Fig. 4d). To do this we designed a completely modular approach with the templating pairs conjugated to the reactive groups and both containing a 17-nt tail designed to hybridize to an antigen-conjugated strand (Fig. 4d). As the recognition element, we employed the p17-peptide, which recognizes HIV-diagnostic antibodies. The templated CuAAC reaction occurred efficiently only in the presence of anti-HIV Ab whereas no product band was detected in its absence or in the presence of other nonspecific antibodies (anti-DIG and anti-DNP antibodies) (Fig. 4e, MS analysis of product shown in Supplementary Fig. 20, for a more detailed interpretation of the bands see Supplementary Fig. 21). In this case, the templated reaction showed an anti-HIV concentration-dependent trend and an efficiency, specificity, and kinetics comparable to those observed with the templated reactions mediated by previous antibodies (Fig. 4f, Supplementary Fig. 22).

**Orthogonal control of reactions mediated by different specific antibodies**. This approach also enables to trigger different templated reactions with different antibodies in the same solution in an orthogonal way. To demonstrate this, we employed in the same solution two templating systems targeting two different antibodies (anti-DIG and anti-DNP). In order to differentiate the products of the two templating reactions, we labeled one of the reactive strands of each templating system with a different fluorophore (FAM and CFR610), so that the final product would be labeled accordingly (Fig. 4g). No product was observed in the absence of both antibodies (Lane 1, Fig. 4h). The addition of one of the two specific antibodies led to the formation of the specific product without any crosstalk with the other templating system (Lanes 2 and 3, Fig. 4h). In the presence of both antibodies, two bands (shown as a single yellow merged band) corresponding to the two products were observed (Lane 4, Fig. 4h).

**Ab-templated synthesis of thrombin-inhibiting aptamer**. To demonstrate the possible utility of our strategy in triggering the synthesis of functional molecules with specific antibodies, we turned our attention to aptamers, single-stranded DNA sequences that are selected in vitro to bind specific targets such as proteins or small molecules, and which might find applications as therapeutic drugs[42]. More specifically, we focused on the thrombin-binding aptamer, a G-quadruplex-forming single-stranded DNA sequence that binds specifically to thrombin, inhibiting its

coagulant function[43]. To demonstrate the Ab-directed synthesis of the thrombin-binding aptamer, we employed a modular approach in which we first split the aptamer into two nonfunctional portions (blue and red strands, Fig. 5a); we then rationally designed at each of the two split ends a 6-nt complementary domain bearing a click-chemistry reactive group at the end (i.e., azide or alkyne) (Fig. 5a). At the opposite end of each portion, we introduced a 17-nt domain that serves to hybridize a DNA strand conjugated with the relevant recognition element (here DIG) (gray strands, Fig. 5a, for a more detailed explanation of the aptamer design, see Supplementary Fig. 23). We observed the formation of the full aptamer only in the presence of anti-DIG Ab (Fig. 5b). The aptamer obtained from the Ab-templated reaction was found to successfully bind to thrombin as demonstrated by native gel electrophoresis (Supplementary Fig. 24). To demonstrate the inhibitory activity of the produced aptamer, a fibrinogen clotting assay was performed in which the thrombin-induced formation of an insoluble fibrin gel was monitored via light scattering measurements[44,45]. To do this, the CuAAC reaction between the aptamer split strands were performed in the absence and presence of anti-DIG Ab and the resulting solution was incubated with thrombin. As expected, the reaction carried out in the absence of anti-DIG Ab did not result in any functional aptamer formation and, as a consequence, we did not observe inhibition activity toward thrombin. Both the thrombin-induced coagulation time and initial coagulation rate were in fact

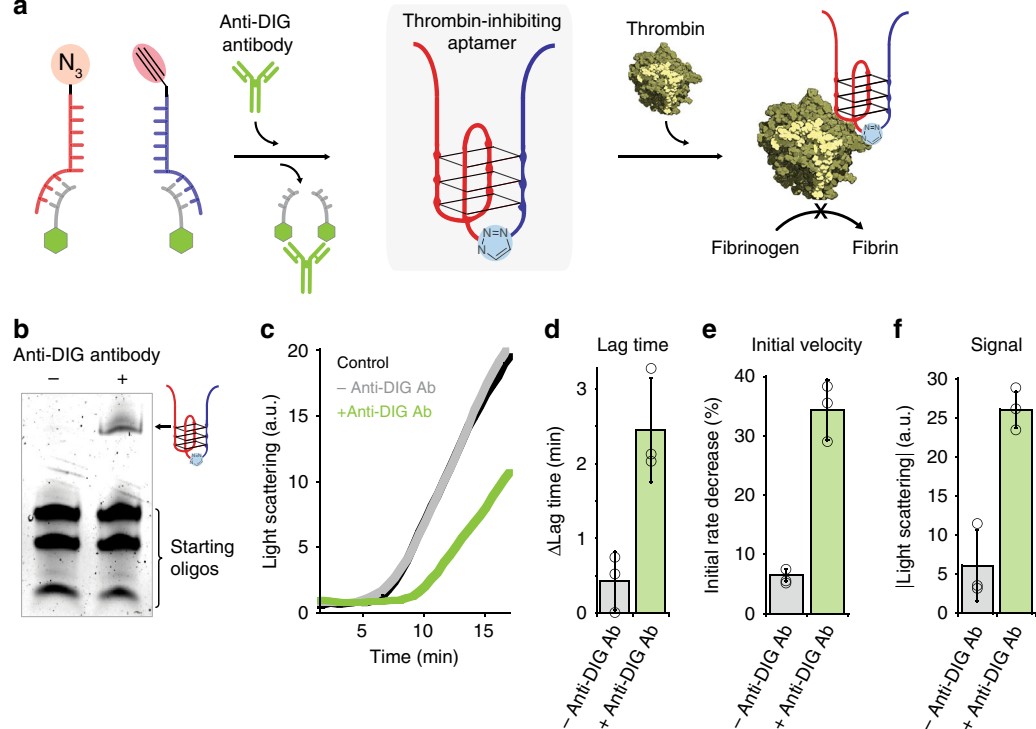

**Fig. 5 Templated synthesis of thrombin-inhibiting aptamer mediated by anti-DIG antibody. a** Scheme of the produced thrombin-inhibiting aptamer by a templated click reaction mediated by anti-DIG antibody. **b** Denaturing PAGE of CuAAC reactions in the absence and presence of anti-DIG antibody. **c** Light scattering measurements of thrombin activity (fibrin formation). Thrombin was incubated with the resulting reaction products obtained from the CuAAC reaction between the antigen-conjugated templating strands in the absence (gray curve) and presence (green curve) of anti-DIG antibody. A control experiment with thrombin alone is also shown as a reference (black curve). **d** Lag time of thrombin activity. Plotted is the difference in the lag time (time required before substrate conversion starts, observed from light scattering measurements) with respect to that of thrombin alone. **e** Reduction of the initial velocity (%) with respect to that of thrombin alone. **f** Maximum light scattering signal observed after 30 min with respect to that of thrombin alone. Templated CuAAC reactions in this figure were all performed at 37 °C for 2 h at a 100 nM concentration of templating strands and 300 nM of anti-DIG antibody in presence of Cu(I) catalyst in 25 mM HEPES buffer pH 7.2, 0.1 M NaCl. Light scattering measurements were accomplished with a 10 mg/ml concentration of fibrinogen, 0.5 nM thrombin, and 125 nM of reaction products in 1% saline solution pH 7 at 25 °C. The experimental values represent averages of three separate measurements and the error bars reflect the standard deviations. Source data are provided as a Source Data file.

indistinguishable from those achieved in the presence of thrombin alone (Fig. 5c). In contrast, the presence of anti-DIG Ab induces the Ab-directed formation of the functional aptamer with thrombin inhibition activity, as demonstrated by the observed longer coagulation lag-time (2.5 min compared to that of thrombin alone, Fig. 5d), by the sharp decrease in initial coagulation rate (34%, Fig. 5e) and by the decrease in light scattering signal observed after 30 min of reaction (Fig. 5f).

## Discussion

DTS represents a powerful and versatile method for controlling the synthesis of different molecules using nucleic acids as templating agents[19]. In recent years, the specificity, selectivity, and programmability of DNA–DNA interactions together with the possibility of conjugating different reactive groups to synthetic oligonucleotides have allowed a wide range of chemical reactions to be directed by a variety of templating architectures[23,46]. The synthesis of different molecules by DTS, including oligomeric and macrocyclic products, has been demonstrated with reactions including ligations, functional group interconversions, and transfer reactions[31]. The advent of synthetic biology together with DNA and aptamer nanotechnology has opened a new route to DTS where nucleic acid templated reactions could be used for targeted drug synthesis triggered by specific nucleic acid biomarkers. Targeted therapy based on nucleic acid-templated synthesis offers many advantages, including overcoming biodistribution and generating different drugs upon recognition of different nucleic acid markers[24].

In this work, we demonstrate that DTS can be controlled by specific protein biomarkers (i.e., IgG antibodies). We rationally engineered different DNA templating strands modified at each end with a pair of reactive groups and suitable recognition elements (i.e., antigen). The bivalent binding of a specific IgG Ab to the so-modified templating strands induces their co-localization and triggers a chemical reaction that would otherwise be impossible at high dilution. We demonstrate that this strategy is versatile and can be easily adapted to different specific IgG antibodies by the simple expedient of changing the recognition event. We also demonstrate that the approach is orthogonal and allows the control of different templated reactions in the same solution by using different triggering antibodies. Our approach takes advantage of the peculiar Y-shaped geometry shared by all IgG antibodies and allows their natural function to be repurposed so that they can act as co-templating agents to control chemical reactivity. Because IgG antibodies represent a wide class of clinical and diagnostic markers, the approach we propose here may prove useful in a range of applications. As a first demonstration of the possible uses of our approach, we show here that this mechanism can be employed to achieve the Ab-directed synthesis of the thrombin-binding aptamer, a molecule with potential therapeutic uses. We demonstrate that the aptamer, produced only in the presence of the specific target Ab, successfully inhibits thrombin coagulation activity, whereas in the absence of the Ab the activity of thrombin is fully retained.

These results, together with the modularity and programmability of our strategy, suggest that similar approaches could be engineered to achieve the targeted synthesis of a wide range of clinically relevant molecules (imaging agents, therapeutic agents, etc.) directly in vivo and in the presence of a specific IgG biomarker Ab. This would represent an avenue to targeted therapy and open horizons in diagnostics and therapeutics.

## Methods

**Chemicals**. Reagent-grade chemicals (HEPES, NaCl, trizma hydrochloride, boric acid, EDTA, urea, acrylamide/bis-acrylamide 40% solution ratio 29:1, APS, TEMED), DIG, copper-catalyzed click reaction reagents (tris(3-

hydroxypropyltriazolylmethyl)amine (THPTA), sodium ascorbate, copper sulfate pentahydrate, and phosphoramidate ligation reagents (1-ethyl-3-(3-dimethylaminopropyl)carbodiimide and 1-(2-hydroxyethyl) imidazole) were purchased from Sigma-Aldrich (St. Louis, Missouri) and used without further purifications. Loading dyes 6× Orange DNA Loading Dye and Gel Loading Buffer II were obtained from ThermoFisher™. DNA-staining dye SYBR® gold was provided by Invitrogen. Sheep polyclonal anti-DIG Ab was purchased from Roche Diagnostic Corporation, Germany, (cat#: 11333089001), mouse monoclonal anti-DNP Ab was purchased from Sigma-Aldrich, USA, (cat#: D8406), murine monoclonal anti-HIV Ab was purchased from Zeptometrix Corporation, USA, (cat#: 0801040) and anti-DIG Fab fragments purchased from Roche Diagnostic Corporation, Germany (cat#: 11214667001). All antibody solutions were aliquoted and stored at a concentration of 1 mg/mL either at 4 °C for immediate use or −20 °C for long-term storage. Human α-thrombin was provided by Hematologic Technologies (Essex Junction, VT). Fibrinogen from human plasma was purchased from Merck (Darmstadt, Germany).

**Oligonucleotides**. Oligonucleotides were purchased HPLC-purified from IBA (Gottingen, Germany), LGC Biosearch Technologies (Risskov, Denmark), or Eurofins (Ebersberg, Germany). All sequences were designed using NUPACK[47]. All oligonucleotides were dissolved in 50 mM PBS pH 7.5 at a 100 μM concentration and stored at −20 °C. Oligo sequences are provided in the Supplementary Information.

**Fluorescence experiments**. Fluorescent experiments were conducted at pH 7.2 in 25 mM HEPES 0.1 M NaCl, at 37 °C in a 100 μL cuvette (total volume of the solution 100 μL). Equilibrium fluorescence measurements were obtained using a Cary Eclipse Fluorimeter with excitation at 550 (±5) nm and acquisition at 570 (±5) nm and data recorded using the commercial Cary Eclipse software (version 1.2). Binding curves were performed using a fixed concentration of 10 nM of Cy3-labeled strand (F-strand#1), and increasing concentrations of Cy5-labeled strand (Q-strand#2), in the absence and in the presence of either Ab-mimic strand (10 nM) or Ab (30 nM). For each concentration, the fluorescence signal in real-time was recorded until it reached equilibrium. The observed fluorescence, $F_{(strand\#1)}$, was fitted using the following four-parameter equation:

$$F_{(strand\#1)} = F_{min} + (F_{max} - F_{min})\frac{K_D}{K_D + [strand\#2]}, \quad (1)$$

where, $F_{min}$ and $F_{max}$ are the minimum and maximum fluorescence values, and $K_D$ is the dissociation constant between templating strands either in the absence ($K_{D\_no\_template}$) or in the presence of the given free concentration of Ab-mimic strand or Ab ($K_{D\_Ab\_mimic}$ or $K_{D\_Anti\_DIG\_Ab}$). The latter apparent dissociation constants are related to $K_{D\_no\_template}$ by the following equation:

$$K_{D\_Ab\_mimic(or\ D\_Anti\_DIG\_Ab)} = K_{D\_no\_template}\frac{K_{Ab\_mimic(or\ Anti\_DIG\_Ab)}}{[Ab\_mimic(or\ Anti\_DIG\_Ab)]}, \quad (2)$$

where $K_{Ab\_mimic}$ or $K_{Anti\_DIG\_Ab}$ are the dissociation constants of the templated duplexes to give the duplex and the free Ab-mimic strand or Ab, respectively. Binding curves allowed the direct evaluation of $K_{D\_no\_template}$ and $K_{D\_Ab\_mimic}$, or $K_{D\_Anti\_DIG\_Ab}$, only for some specific lengths of the complementary domains of templating pairs. However, judicious assumptions allowed reliable estimates of the constants not amenable to direct evaluation for all the other cases (see SI).

**Kinetic model**. In order to evaluate the yields as a function of reaction parameters, the reaction between templating pairs in the absence and in the presence of either Ab-mimic strand or anti-DIG Ab was modeled by the simple kinetic scheme shown in Fig. 1. In this scheme, the fast and reversible formation of a duplex, characterized by the dissociation constant $K_D$, is followed by the irreversible reaction between the co-localized end groups, characterized by the kinetic constant $k$. The dissociation constant $K_D$ is meant as the dissociation constant between templating strands either in the absence ($K_{D\_no\_template}$) or in the presence of the given free concentration of Ab-mimic strand or Ab ($K_{D\_Ab\_mimic}$ or $K_{D\_Anti\_DIG\_Ab}$). Integration of the corresponding rate equation allows obtaining the *reaction yield* (%) by the following equation (see SI):

$$\text{Reaction yield}(\%) = \left[c_0 - \frac{K_D}{2\omega(z)}\left(\frac{1}{2\omega(z)} + 1\right)\right]\frac{100}{c_0}, \quad (3)$$

where $c_0$ is the initial concentration of the two strands, and $\omega(z)$ is the Wright omega function, whose argument, $z$, is given by Eq. (4), where $t$ is the reaction time:

$$z = \frac{1}{2}kt + \left[(1 + 4c_0/K_D)^{1/2} - 1\right]^{-1} - \ln\left[(1 + 4c_0/K_D)^{1/2} - 1\right]. \quad (4)$$

In Figs. 2g and 3e has reported a plot of the reaction yield vs. $1/K_D$ calculated by Eq. (3) under the conditions $c_0 = 10^{-7}$ M, $k = 0.465$ h$^{-1}$, $t = 2$ h. The value of $k$ was optimized so as to give the best fit to the experimental yields reported in both Figs. 2g and 3e.

**Ab-mimic strand-directed CuAAC reaction**. The 5′-azide modified (strand#1), 3′-alkyne modified (strand#2), and unmodified (Ab-mimic strand) oligos were

combined at a concentration of 100 nM in a buffer containing 25 mM HEPES pH 7.2; 0.1 M NaCl. The reaction was initiated by the addition of 1 µl from a copper(l) catalyst solution containing THPTA ligand (final 117 µM), sodium ascorbate (final 167 µM) and $CuSO_4 \cdot 5H_2O$ (final 16.6 µM), and left 2 h at 37 °C without shaking. The total volume of the reaction was 10 µl.

**Anti-DIG and anti-DNP Ab-directed CuAAC reaction.** The 5′-azide modified (strand#1), 3′-alkyne/5′-antigen modified (antigen-strand#2), and 5′-antigen modified (antigen-strand#3) oligos were combined at a concentration of 100 nM in a buffer containing 25 mM HEPES pH 7.2; 0.1 M NaCl. The corresponding Ab was added to the solution at the concentration indicated for each case and left for 15 min to allow for stabilization of the complex Ab-oligos. The reaction was initiated by the addition of 1 µl from a copper(l) catalyst solution containing THPTA ligand (final 117 µM), sodium ascorbate (final 167 µM) and $CuSO_4 \cdot 5H_2O$ (final 16.6 µM), and left 2 h at 37 °C without shaking. The total volume of the reaction was 10 µl.

**Anti-DIG Ab-directed phosphoramidate ligation.** The 5′-amino-modified ($NH_2$-strand#1), 3′-phosphate/5′-DIG modified (Phos-DIG-strand#2) and 5′-DIG modified (DIG-strand#3) oligos were combined at a concentration of 100 nM in a buffer containing 25 mM HEPES pH 7.2; 0.1 M NaCl. Anti-DIG Ab was added to the solution at the concentration indicated and left for 15 min. The reaction was initiated by the addition of EDC·HCl (final concentration 25 mM) and 1-(2-hydroxyethyl) imidazole (final concentration 100 mM) and left 2 h at 37 °C without shaking. The total volume of the reaction was 10 µl.

**Anti-HIV Ab-directed CuAAC reaction.** The 5′-azide modified (strand#1), 3′-alkyne modified (HIV-strand#2) were added at a concentration of 100 nM and the 5′-peptide modified PNA sequence (HIV-strand#3) was added at a concentration of 200 nM in a buffer containing 25 mM HEPES pH 7.2; 0.1 M NaCl. Anti-HIV Ab was added to the solution at the concentration indicated and left for 15 min. The reaction was initiated by the addition of 1 µl from a copper(l) catalyst solution containing THPTA ligand (final 117 µM), sodium ascorbate (final 167 µM), and $CuSO_4 \cdot 5H_2O$ (final 16.6 µM), and left 2 h at 37 °C without shaking. The total volume of the reaction was 10 µl.

**Orthogonal control of CuAAC reactions by different antibodies.** Solutions of oligo duplexes at a concentration of 1 µM were prepared beforehand and submitted to annealing (heating at 95 °C for 5′ followed by slow cool down for 30′). The 5′-azide/3′-FAM modified (FAM-strand#1) and 5′-DIG (DIG-strand#3) duplex and the 5′-azide/3′-CFR610 modified (CFR610-strand#1) and 5′-DNP (DNP-strand#3) duplex were added together with single oligos 3′-alkyne/5′-DIG (DIG-strand#2) and 3′-alkyne/5′-DNP (DNP-strand#2) at a concentration of 100 nM in a buffer containing 25 mM HEPES pH 7.2; 0.1 M NaCl. Anti-DIG and anti-DNP antibodies were added to the solution at a concentration of 300 nM when it applies and left for 15 min. The reaction was initiated by the addition of 1 µl from a copper(l) catalyst solution containing THPTA ligand (final 117 µM), sodium ascorbate (final 167 µM) and $CuSO_4 \cdot 5H_2O$ (final 16.6 µM), and left 2 h at 37 °C without shaking. The total volume of the reaction was 10 µl.

**Thrombin-inhibiting aptamer formation mediated by anti-DIG Ab.** Both aptamer splits (split#1 and split#2) together with the two corresponding DIG-modified oligos (DIG-strand#3 and DIG-strand#4) were combined at a concentration of 100 nM in a buffer containing 25 mM HEPES pH 7.2; 0.1 M NaCl. Anti-DIG Ab was added at a 300 nM concentration and left for 15 min to allow for stabilization of the complex Ab-oligos. The reaction was initiated by the addition of 1 µl from a copper (l) catalyst solution containing THPTA ligand (final 117 µM), sodium ascorbate (final 167 µM) and $CuSO_4 \cdot 5H_2O$ (final 16.6 µM), and left 2 h at 37 °C without shaking. The total volume of the reaction was 10 µl.

**PAGE experiments.** Native gels (8%) were run at room temperature in running buffer (1× TBE, pH 8) at 100 V. Sample volumes of 10 µl were combined with 1 µl of 6× Orange DNA Loading Dye followed by direct loading into the gel. Denaturing polyacrylamide gels (18%) were prepared at a concentration of 7 M urea and run at room temperature in running buffer (1× TBE, pH 8) at 150 V. A sample volume of 10 µl was combined with 1 µl of Gel Loading Buffer II and 3 µl of formamide and then submitted to heat treatment at 95 °C for 5 min followed by a fast cooling in an ice-bath for 3 min. The experiments were carried out in a Mini-PROTEAN Tetra cell electrophoresis unit (Bio-Rad) using Bio-Rad PowerPac Basic power supply. If appropriate, gels were first scanned by a ChemiDoc MP imaging system (Bio-Rad) to detect fluorophores (FAM and CFR610), then stained with SYBR Gold and imaged. Reaction yields were estimated based on band intensities by densitometry using Image Lab software from Bio-Rad (version 6.0.1).

**Thrombin inhibitory tests by light scattering measurements.** A 10 µl solution containing thrombin (final concentration: 0.5 nM) and the resulting reaction products obtained from the aptamer CuAAC reaction formation between the antigen-conjugated templating strands (in the presence or absence of anti-DIG Ab,

as indicated) previously EtOH precipitated (final concentration of strands: 125 nM) was incubated for 10 min at 4 °C. Then, the mixture was added to 50 µl of a 10 mg/ml solution of fibrinogen in 1% NaCl. The enzymatic reaction was followed by the light scattering caused by the formation of insoluble fibrin at 25 °C in a 45 µL cuvette (total volume of the solution 60 µL) by using a Cary Eclipse Fluorimeter with excitation at 600 (±10) nm and acquisition at 610 (±10) nm.

**UPLC and mass spectrometry.** Ultra-performance liquid chromatography–MS (UPLC-MS) analysis of oligonucleotides was performed on a Waters Acquity H Class system coupled to a Waters Xevo G2-XS QToF mass spectrometer. Chromatography was carried out on a Waters Acquity BEH C4 column (1.7 µm particle size, 2.1 × 50 mm) at a flow rate of 0.3 mL/min using a gradient of buffers A and B: buffer A, 400 mM 1,1,1,3,3,3-hexafluoroisopropanol, 15 mM triethylamine; buffer B, 100% methanol. The gradient was 10–45% buffer B over 15 min. The eluent was injected directly into the mass spectrometer, and the data acquired in the negative-ion mode (ESI−, mass range: 500–3000 m/z). Data were analyzed and deconvoluted by using the manufacturer's software (MaxEnt1, UNIFI, Waters).

**Statistics and reproducibility.** Gel electrophoresis experiments have been replicated three times independently to ensure reproducibility of results.

**Reporting summary.** Further information on research design is available in the Nature Research Reporting Summary linked to this article.

## Data availability

Data are available upon request from the corresponding author. Source data are provided with this paper.

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

## Acknowledgements

This work was supported by Associazione Italiana per la Ricerca sul Cancro, AIRC (project n. 21965) (F.R.), by the European Research Council, ERC (Consolidator Grant project n. 819160) (F.R.) and by the Marie Skłodowska-Curie ITN project DNA-robotics (project n. 765703) (F.R., L.B.P., and T.B.).

## Author contributions

L.B.P. and F.R. conceived and designed the experiments; L.B.P., M.M., and F.R. performed the experiments and analyzed the data; J.W. and T.B. performed the characterization of the products; G.E. designed the kinetic model; L.B.P., G.E., and F.R. wrote the paper.

## Competing interests

The authors declare no competing interests.
