## [Peer Review File · Nature Communications]

Reviewers' Comments:

Reviewer #1:

Remarks to the Author:

Ricci and co-workers describe an elegant, rational approach for DNA-templated synthesis controlled by antibodies. The work has been done with great care and the authors employ a detailed thermodynamic and kinetic study on this phenomenon which is supported by extensive quantitative modeling. The work is very impressive, and the scholarly presentation is high. I suggest the work can be accepted after the following minor issues are resolved.

Main comments

- 1) I believe that the drop in the reaction yield vs concentration of [Ab] observed at high concentration is indicative of what is called combinatorial inhibition, first described by Perelson in 1981 (Receptor clustering on a cell surface. II. theory of receptor cross-linking by ligands bearing two chemically distinct functional groups). This is a well-known effect in ternary equilibria further worked out in <https://pubs.acs.org/doi/10.1021/ja311795d> and <https://www.pnas.org/content/97/11/5818>. However, the concentration where this dip occurs should depend on the K_d of the interaction between the Ab and antigen. However, in all your curves it occurs at the same concentration which is rather strange as I think the K_d values are quite different. Could the authors comment on this? It would also be good to have a table where all the K_d values of the different antigen-antibody interactions is specified.
- 2) In all the plots of reaction yield vs [AB-mimic] or [Anti-X Ab], the point just before the decrease in reaction yield has the highest error. Can the authors comment why this is? Also, most other error bars in these plots are not visible. Are they plotted or just too small?

Minor comments

- 1) Figures captions in general. Please specify the temperature at which the measurements have been done. Currently, for some measurements like 2h the temperature is given in the caption but for other experiments (like 2b) no temperature is given.
- 2) Figure 2c: The caption belonging to this figure is incomplete and should also contain the ratio and concentration of components. This is specified in the main text but should also be done in the caption. Figure S17 has the same problem. In general, try to be as complete as possible in a Figure caption.
- 3) Abstract: "DNA templating strands that triggers an intermolecular reaction..". I could be wrong here but is the templated reaction not intramolecular?
- 4) Derivation of eq S30 in the SI. The authors state: "Since the percent reaction yield is defined as $(c_0 - c_t)100/ c_0$, from equation (S29) the reaction yield can be easily calculated as a function of c_0 , $KD_no_template$, k , and t , by equation (S30). However, eqS30 does not explicitly depend on time. I think the authors also need to mention eq4 in the main text here which shows how z depends on t .

Reviewer #2:

Remarks to the Author:

The paper by Pellejero and colleagues demonstrated a strategy for the control of DNA-templated chemical reactions using specific IgG antibodies as co-templating agents. This approach is based on the co-localization of reactants induced by the bivalent binding of a specific IgG antibody to two antigen-conjugated DNA templating strands that triggers an intermolecular reaction. The strategy is versatile, orthogonal and can be easily adapted to different specific IgG antibodies by the simple expedient of changing the recognition event. They also reported the antibody-directed synthesis of a thrombin-inhibiting aptamer and prove successful inhibition of thrombin coagulation activity only in the presence of a specific IgG antibody.

This is a promising strategy that takes advantage of both the sequence-specificity of DNA-DNA interactions and the possibility of using nucleic acids as versatile scaffolds to conjugate reactive

groups and different recognition molecules. This paper has excellent novelty and advantages in the field of synthetic biology, and is of great value for publication.

I suggest this manuscript be published after the following minor revisions:

- (1) Language, grammars, and figures should be further polished.
- (2) "The modularity and programmability of our strategy suggests that ..." should be "The modularity and programmability ...suggest..." on line 34 in page 2.
- (3) "and, second, the possibility of..." on line 99 in page 5, where there are two conjunctions, which should be changed to "second, the possibility of...".

Francesco Ricci
Professor
University of Rome, Tor Vergata
Via della Ricerca Scientifica, 1, 00133, Rome, Italy
Email: francesco.ricci@uniroma2.it
www.francescoricci.com

October 28, 2020

RE: Manuscript title: Using antibodies to control DNA-templated chemical reactions

Response to reviewers:

We detail below the responses to the reviewers' comments and the changes we have made to the manuscript in response to the reviewers' suggestions.

Reviewer #1:

Main comments

1) I believe that the drop in the reaction yield vs concentration of [Ab] observed at high concentration is indicative of what is called combinatorial inhibition, first described by Perelson in 1981 (Receptor clustering on a cell surface. II. theory of receptor cross-linking by ligands bearing two chemically distinct functional groups). This is a well-known effect in ternary equilibria further worked out in <https://pubs.acs.org/doi/10.1021/ja311795d> and <https://www.pnas.org/content/97/11/5818>.

However, the concentration where this dip occurs should depend on the K_d of the interaction between the Ab and antigen. However, in all your curves it occurs at the same concentration which is rather strange as I think the K_d values are quite different. Could the authors comment on this? It would also be good to have a table where all the K_d values of the different antigen-antibody interactions is specified.

We thank the Reviewer for this insightful comment that made us better consider our system. The reviewer suggests that the drop in the reaction yield vs [Ab], observed at high concentration of Ab, is indicative of what is called combinatorial inhibition, an effect that has been described in the literature of ternary complexes. He/she further suggests that the concentration where this drop occurs should depend on the K_d of the interaction between the Ab and antigen. The reviewer is correct: the mechanism we proposed for the observed drop, illustrated in Supplementary Figure 8, is indeed analogous to the mechanism of combinatorial inhibition in classical ternary complexes. There is, however, one notable difference that could be the reason for the behavior we observe: in our case the binding of an additional Ab molecule opens a ring structure as evidenced in the scheme below:

In this process, an intramolecular interaction is substituted by an analogous intermolecular interaction, and the equilibrium constant is dictated by the effective molarity (EM) of the ring being disrupted rather than the K_d of the interaction between the Ab and antigen. Should this difference be sufficient to justify the fact that the observed drop in our curves occurs at concentrations of the same order of magnitude? Actually, it is hard to say. The observed drop in our work, while surely interesting to better understand the system, was a marginal observation that we did not investigate any further and did not try to model. It is important, however, to remark that our kinetic modelling was carried at a fixed concentration of Ab-mimic strand, and Anti-DIG antibody, where the effect of combinatorial inhibition can be safely neglected. As to the values of K_d of the different antigen-antibody interactions, actually we determined by titration only the K_d value for the interaction of the antigen with the anti-DIG antibody ($K_{Anti_DIG_Ab} = 3.6 \times 10^{-9}$ M; see section in SI titled: Analysis of the binding curves in the presence of the anti-DIG antibody), and the K_d value for the interaction of the DNA-duplex with the Ab-mimic strand ($K_{Ab_mimic} = 2.2 \times 10^{-10}$ M; see section in SI titled: Analysis of the binding curves in the presence of the Ab-mimic strand). From these values and the values of the dissociation constants for DNA duplexes of different complementary lengths, reported in Table S2, we calculated the values of the apparent dissociation constants for DNA templated duplexes of different complementary lengths in the presence of $[Anti_DIG\ antibody] = 10^{-7}$ M (reported in Table S5), and the values of the apparent dissociation constants for DNA templated duplexes of different complementary lengths in the presence of $[Ab_mimic] = 10^{-7}$ M (reported in Table S3).

2) In all the plots of reaction yield vs $[AB\text{-mimic}]$ or $[Anti\text{-X Ab}]$, the point just before the decrease in reaction yield has the highest error. Can the authors comments why this is? Also, most other error bars in these plots are not visible. Are they plotted or just too small?

The error is shown for the point that leads to the highest reaction yield, and thus the one which is thoroughly used within the work (1×10^{-7} M for $[Ab\text{-mimic}]$ and 3×10^{-7} M for $[Anti\text{-X Ab}]$). We thank the reviewer for this comment that allows us to better clarify an important aspect of our experiments: the values of reaction yields are obtained by densitometric analysis of the bands from gel electrophoresis. Gel electrophoresis obviously does not represent the best technique to achieve quantitative results of yield but does a good job in giving a precise picture of the reactivity trend, which is the one we are interested in emphasizing in these plots. We appreciate that the reviewer pointed out this and we have revised the caption of the plots accordingly for clarification.

Minor comments

1) Figures captions in general. Please specify the temperature at which the measurements have been done. Currently, for some measurements like 2h the temperature is given in the caption but for other experiments (like 2b) no temperature is given.

We thank the reviewer for this comment. The temperature at which the experiments were performed is now added in the text.

2) Figure 2c: The caption belonging to this figure is incomplete and should also contain the ratio and concentration of components. This is specified in the main text but should also be done in the caption. Figure S17 has the same problem. In general, try to be as complete as possible in a Figure caption.

Following the reviewer comment we have improved the captions' description.

3) Abstract: "DNA templating strands that triggers an intermolecular reaction..". I could be wrong here but is the templated reaction not intramolecular?

The referee is right. The reaction occurs between two separate strands that are brought together (and hybridize) through the bivalent binding of an antibody. To avoid any confusion we have revised this sentence as: "Our approach is based on the co-localization of reactants induced by the bivalent binding of a specific IgG antibody to two antigen-conjugated DNA templating strands that triggers a chemical reaction that would be otherwise too slow to have an observable effect under diluted conditions".

4) Derivation of eq S30 in the SI. The authors state: "Since the percent reaction yield is defined as $(c_0 - ct)100/ c_0$, from equation (S29) the reaction yield can be easily calculated as a function of c_0 , $KD_no_template$, k , and t , by equation (S30). However, eqS30 does not explicitly depend on time. I think the authors also need to mention eq4 in the main text here which shows how z depends on t .

Equation 4 in the main text has been mentioned in the SI as the new equation (S31).

Reviewer #2:

Reviewer #2 was very positive about our work and suggested minor revisions before acceptance.

(1) Language, grammars, and figures should be further polished.

We thank the reviewer for this suggestion. We have further polished language, grammars and the figures to avoid typos and for a better clarity of the paper.

(2) "The modularity and programmability of our strategy suggests that ..." should be "The modularity and programmability ...suggest..." on line 34 in page 2.

We thank the reviewer for spotting this typo. We have revised the text accordingly.

(3) "and, second, the possibility of..." on line 99 in page 5, where there are two conjunctions, which should be changed to "second, the possibility of..."

We thank the reviewer for spotting this typo. We have revised the text accordingly.

Reviewers' Comments:

Reviewer #1:

Remarks to the Author:

The authors have carefully addresses all the points that I raised. As such the manuscript can be accepted in its current form. I want to congratulate the authors with such a fine piece of work.